# Genetic Screening Reveals Heterogeneous Clinical Phenotypes in Patients with Dilated Cardiomyopathy and Troponin T2 Variants

**DOI:** 10.3390/jpm13040611

**Published:** 2023-03-31

**Authors:** Angelika Weis, Svenja Krueck, Gregor Dombrowsky, Anne Schänzer, Christian Jux, Anselm Uebing, Inga Voges, Marc-Phillip Hitz, Stefan Rupp

**Affiliations:** 1Department of Pediatric Cardiology, Intensive Care Medicine and Congenital Heart Disease, Justus Liebig University Giessen, 35390 Giessen, Germany; 2Department of Congenital Heart Disease and Pediatric Cardiology, University Hospital of Schleswig-Holstein, 24105 Kiel, Germany; 3Institute of Medical Genetics, Carl von Ossietzky University, 26129 Oldenburg, Germany; 4Institute of Neuropathology, Justus Liebig University Giessen, 35390 Giessen, Germany; 5DZHK (German Centre for Cardiovascular Research), Partner Site Hamburg/Lübeck/Kiel, 24105 Kiel, Germany

**Keywords:** troponin T2, *TNNT2*, cardiomyopathy, dilated cardiomyopathy, genetic screening

## Abstract

Background: Cardiomyopathies (CMs) are a heterogeneous and severe group of diseases that shows a highly variable cardiac phenotype and an incidence of app. 1/100.000. Genetic screening of family members is not yet performed routinely. Patients and methods: Three families with dilated cardiomyopathy (DCM) and pathogenic variants in the troponin T2, Cardiac Type (*TNNT2*) gene were included. Pedigrees and clinical data of the patients were collected. The reported variants in the *TNNT2* gene showed a high penetrance and a poor outcome, with 8 of 16 patients dying or receiving heart transplantation. The age of onset varied from the neonatal period to the age of 52. Acute heart failure and severe decompensation developed within a short period in some patients. Conclusion: Family screening of patients with DCM improves risk assessment, especially for individuals who are currently asymptomatic. Screening contributes to improved treatment by enabling practitioners to set appropriate control intervals and quickly begin interventional measures, such as heart failure medication or, in selected cases, pulmonary artery banding.

## 1. Background

Cardiomyopathies (CMs) are a heterogeneous but severe group of diseases that involve the heart phenotypically and functionally, with an incidence of app. 1/100,000 in the pediatric age group. They are classified according to their phenotype, with dilated cardiomyopathy (DCM) representing the largest proportion (approximately 50%), followed by hypertrophic cardiomyopathy (HCM) in approximately 40% of cases, while rarer forms include restrictive cardiomyopathy (RCM), arrhythmogenic ventricular cardiomyopathy (AVC), and noncompaction cardiomyopathy [1].

The subclassification of CM is complex and not inherent. This is partly due to different underlying etiologies resulting in CM. Most CMs in children are genetically determined and only affect the heart. These CMs are designated as primary CM. However, CM might also arise in combination with complex diseases involving the skeletal muscle or other organs [2].

Genetic testing in pediatric cardiomyopathy cases is a valuable tool for diagnosis and classification. Due to the increasing use of genetic testing, the detection rate of pathological causative genetic variants has increased up to 60% [2,3]. However, genetic testing is not yet performed routinely.

Genes causing primary CM appear to encode proteins that frame a “final common pathway” for that specific disorder. The “final common pathways” for the classified forms of CM include the sarcomere in the primarily diastolic dysfunction disorders HCM and RCM, the linkage of the sarcomere and sarcolemma in the systolic dysfunction disorder DCM, and the desmosome in AVC [4].

Disease manifestation can vary, for reasons not yet understood, with regard to phenotypic characteristics, age of onset, disease progression, and eventual outcome. Whether adopting genetic diagnostics in CM is useful in routine diagnostics is explored in this article, which reports three families with primary CM.

### Troponin T2, Cardiac Type (TNNT2)

*TNNT2* codes for the cardiac isoform of the cardiac troponin T subunit. *TNNT2* binds to tropomyosin and helps position it on actin, and together with the rest of the troponin complex, modulates contraction [5]. Variants in *TNNT2* can lead to different CM phenotypes, such as HCM [6,7], RCM [8], and DCM [6,9,10]. In one study, the age of onset for DCM caused by a *TNNT2* variant was 16–45 years in 4 families (13 patients) [10]. In another study, the age of onset differed more significantly [9]. This family’s pedigree was extended and included in the present study.

## 2. Materials and Methods

### 2.1. Patients

Three families were included, hereafter referred to as families A, B, and C. The index patients from families A and C were admitted to the pediatric cardiology department for heart transplantation, and index patient B was admitted with mild symptoms of heart failure. Clinical data for the patients are provided in Table 1. For the index patient in family A, a genetic panel test, including the genes *LDB3*, *LMNA*, *MYH7*, *MYBPC3,* and *TNNT2,* was performed. In families B and C, an expanded panel-based genetic test was conducted in the index patient. Genetic testing for the identified variant was thereafter performed in all first-degree relatives. The rating of variants was performed according to American College of Medical Genetics (ACMG) guidelines [11]; see also Appendix A. Variants were confirmed in first-degree family members by Sanger sequencing. Pedigrees were drawn according to standard recommendation [12,13].

### 2.2. Cardiac Histology

Heart muscle tissue was obtained from the left ventricle of the explanted heart from patient A.III.2 at the age of 13 years. From formalin-fixed paraffin-embedded (FFPE) tissue, H&E and Masson trichrome staining and immunohistochemistry were performed, utilizing antibodies against N-Cadherin (mouse monoclonal, Abcam). Small biopsies were fixed with 6% glutaraldehyde and embedded in epoxy resin. Semithin sections were stained with Richardson’s. For transmission electron microscopy (TEM), ultrathin sections were contrasted with 3% lead citrate-3H2O.

## 3. Results

### 3.1. Family A

A two-year-old female patient (A.III.3) with DCM with non-compaction was admitted for heart transplantation. The parents were not related. Family history revealed that three of the father’s (A.II.2) sisters (A.II.4-A.II.6) died in infancy due to cardiac failure. The 22-year-old father (A.II.2) and the paternal 52-year-old grandmother (A.I.1) were also diagnosed with DCM. Both were asymptomatic at the time of diagnosis [9]. Genetic testing revealed a likely pathogenic *TNNT2* variant NM_001276345.2(*TNNT2*):c.316G>A (p.Glu106Lys) in the index patient’s father and paternal grandmother. The index patient presented at the age of two months with clinical signs of heart failure that subsequently worsened.

The patient underwent heart transplantation at the age of 26 months. In sibling A.III.2, genetic diagnosis after birth revealed the same variant. He was regularly seen in the outpatient clinic and was asymptomatic with normal echocardiographic finding until the age of 13 years. He deteriorated rapidly within six months and received a heart transplant. Histological findings of this patient are provided in Figure 1. Histological analysis of left ventricular tissue from the explanted heart revealed a cardiomyopathy with moderate cellular hypertrophy, in addition to increased variation of cardiomyocytes and mild myocardial fibrosis. A disarrangement of myofibril architecture and sarcomere structure was present, with irregularity of the intercalated disks. Mitochondrial aggregation of swollen and dissolved mitochondria was observed with some unspecific lipofuscin deposits. In a third male sibling (A.III.1), the pathogenic *TNNT2* variant was also detected. He is seen for regular cardiac follow-ups and has no clinical or echocardiographic signs of CM at the age of five years (pedigree in Figure 2).

### 3.2. Family B

In this family, an infant (B.III.3) presented with severe DCM and required a heart transplant at 8 months of life. At the age of 12 years, he died after a rejection event and subsequent re-transplantation. Several years later, his brother (B.III.4) presented to the pediatric emergency department with a cough and tachycardia. He was diagnosed with severe DCM and died a few weeks later at the age of 12 years. His sister (B.III.5) underwent regular cardiovascular follow-ups for screening and was diagnosed with DCM during early adolescence with a decline in LV function within a few years. An in-frame deletion NM_001276345.2(*TNNT2*):c.659_661del p.(Lys220del) in *TNNT2* was detected in her. She underwent heart transplantation and passed away several years after transplantation due to myocardial infarction. Their half-brother (B.III.6) carries the same pathogenic *TNNT2*-variant and was diagnosed with mild LV dilatation in early childhood. His latest echocardiography and cardiovascular magnetic resonance (CMR) imaging findings show mild LV impairment with an EF of 51% and a mild increase of LV volumes. The father of the four patients (B.II.3) was also affected by DCM, and he died at the age of 50 years due to unknown reasons. The genetic constellation within the family suggests that he was an obligate carrier of the *TNNT2* variant detected in his offspring (pedigree in Figure 3).

### 3.3. Family C

A 1 ½-year-old toddler (C.II.1) was admitted for heart transplant. The patient initially presented with vomiting and feeding difficulties. At the first presentation in a peripheral hospital, obstructive bronchitis was suspected. A chest X-ray showed cardiomegaly and subsequent echocardiography revealed a DCM. A likely pathogenic variant in the *TNNT2* gene (NM_001276345.2(*TNNT2*):c.644G>C p.(Arg215Pro)) was identified. In addition to acute anti-congestive therapy (milrinone, bisoprolol, HCT, furosemide, and levosimendan once per week), the patient was listed for heart transplantation. However, at the same time (one month after admission), pulmonary arterial banding was performed [14]. This led to a significant improvement in the patient’s condition and, during follow-up, a partial balloon dilatation of the banding was performed. Twelve years after initial pulmonary artery banding, the patient is in good clinical condition (NHYA I, EF of 50%) and still receiving oral heart failure medication (bisoprolol, lisinopril, and spironolactone). CMR images before and after pulmonary artery banding are shown in Figure 4.

Genetic testing was also performed on first-degree relatives. The pathogenic variant was detected in her mother (C.I.2) as well as in her sister (C.II.2). The mother currently shows moderate functional left ventricular impairment (EF 45%). The sister (C.II.2) has mild functional left ventricular impairment on echocardiography (EF 50%). (pedigree in Figure 5)

## 4. Discussion

In the reported patients with variants in *TNNT2*, DCM occurred in a familial form with autosomal dominant inheritance and penetrance of 94% or higher, as the patients without penetrance were at a younger age at the most recent follow up. In contrast to the literature, only DCM was seen, with one case showing additional features of non-compaction.

Histological analysis of tissue from the left ventricle of the explanted heart in one patient of our study revealed the common DCM phenotype with moderate cellular hypertrophy, mild myocardial fibrosis, irregularity of the intercalated disks, increased variation of cardiomyocytes, disarrangement of myofibril architecture and sarcomere structure, and aggregation of swollen and dissolved mitochondria with the occurrence of lipofuscin.

In total, we reported 16 patients from 3 families with an overall poor outcome of DCM due to variants in *TNNT2*. Twelve of these patients were children. Eight patients died or received a heart transplantation. Four patients died with signs of heart failure without prior genetics. The high number of patients with a poor outcome emphasizes the severity of the identified variants in the *TNNT2* gene. The age of onset of CM ranged from the perinatal period to manifestation in the fifth decade.

Of additional interest, one child, at 13 years of age, deteriorated from NYHA stage I to NHYA stage IV, and underwent heart transplantation within six months. This child was not treated with anti-congestive therapy due to a normal clinical condition and echocardiography until acute decompensation occurred, although the variation was known.

Autosomal dominant variants in *TNNT2* are a well-established genetic cause for dilated, hypertrophic, restrictive, or non-compaction CM [6]. However, the clinical phenotype of *TNNT2*-associated CM shows high variability, as illustrated by an overview of clinically reported variants in *TNNT2* [Appendix A]. Of the 58 variants included in this overview, 31 variants (53%) are reported in the context of hypertrophic CM, 27 (47%) in non-compaction CM, and 22 (38%) in restrictive CM. Only 12 variants (21%) are related to dilated CM. No clear correlation between variant location within the protein and corresponding phenotype can be observed [Appendix A]. In fact, different phenotypic outcomes are observed even for closely colocalized variants or carriers of identical variants (Appendix A). The variant reported in family B (Clinvar ID 43659) can serve as a representative example for this observation, as it has been observed in patients with dilated cardiomyopathy, hypertrophic cardiomyopathy, left ventricular noncompaction, or restrictive cardiomyopathy. Conversely, the affected individuals presented herein mainly differed by age of onset and severity, but consistently exhibited a DCM phenotype. This variant results in the deletion of one lysine in a motif of four lysines. Previous publications have generated functional evidence that this variant decreases the Ca^2+^ sensitivity and results in reduced inter-troponin interactions [10,11]. Taken together, the evidence for pathogenicity of this alteration is clear, although the causes for the different phenotypic outcomes remain unresolved. The variant reported in family A was previously described by Luedde et al. [9]. This publication also presented functional data including a transgenic mouse model, which showed a DCM phenotype, further strengthening the association presented here. To our knowledge, the variant in family C has not been described in a previous publication. However, an entry in Clinvar with conflicting interpretations of pathogenicity exists (Clinvar ID 229338). One of the submissions reports the variant in a patient with DCM but no detailed information was given. Colocalized variants have been reported as (likely) pathogenic (p.Arg215Trp Clinvar ID 180554; p.Arg215Leu Clinvar ID 12416), which further underlines the importance of this residue. For all three variants, the affected amino acid residue is evolutionarily well conserved (Appendix A). Overall, not only does the cardiac phenotype of *TNNT2*-associated CM vary [6,8,9,10], but the age at disease onset also shows a large range. Furthermore, the differences observed even in carriers of identical genetic variants suggest the influence of modifiers that promote the onset and development of a specific phenotype. Those modifiers might include environmental factors, such as exercise [15], and additional genetic influences, such as polygenic risk factors or an additional trigger through an infectious agent, e.g., myocarditis [16].

We know from other studies that DCM leads to a high degree of myocardial fibrosis in older patients [17,18]. In contrast, fibrosis only plays a minor role in very young patients, and the mRNA expression in the heart varies significantly in different age periods [17]. These findings may offer the possibility of adopting new therapeutic options. While all patients with congestive heart failure should be treated with anti-congestive medication, a new therapy has already evolved for patients under the age of 2 years and preserved RV function [14]. A multicenter study reporting 70 patients described improved function in 70% of patients and a recovery in approximately half of the patients [14].

The study presented here illustrates the benefits of genetic screening in familial CM. Standard genetic testing can help identify pathogenic variant carriers, who can subsequently undergo cardiac screening and, if appropriate, receive anti-congestive treatment sooner. It must be discussed how often children with known pathogenic *TNNT2* variants or similar CM-associated genes should be seen by a pediatric cardiologist, and when anti-congestive treatment should be initiated.

The genetic basis of CM is a field developing at an extraordinary pace. It is not yet understood why the same variant can lead to DCM in one case and HCM in another. Likewise, it is unclear why the age of disease manifestation and its severity varies enormously, as presented here, even for carriers of an identical variant. Especially on the molecular level, exciting new results can be expected, which may provide insights into the underlying mechanisms and potentially lead to improvements for the patient or introduce variant-specific treatments.

## Figures and Tables

**Figure 1 jpm-13-00611-f001:**
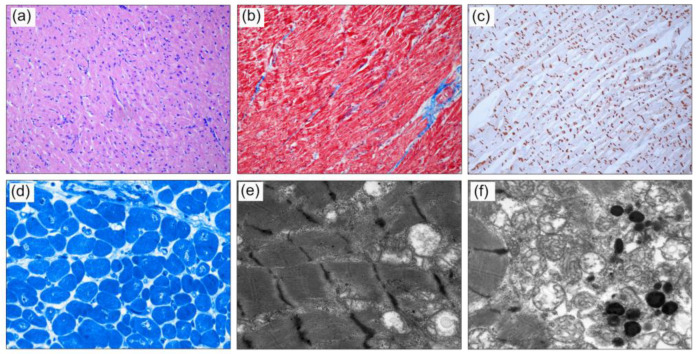
Histology of the left ventricle of the explanted heart from patient A.III.2: (**a**) H&E-stained sections show moderate myocardial cellular hypertrophy. (**b**) Mild myocardial fibrosis (blue) is visualized with Masson Trichrome. (**c**) Immunohistochemistry with antibodies against N-Cadherin shows focal irregularity of the intercalated disks (IDs) in longitudinal sections. (**d**) On Richardson’s-stained resin sections, increased variation of cardiomyocytes and enlarged nuclei, partly with irregular shape, are present. (**e**) Representative TEM images showing disarrangement of myofibril architecture and sarcomere structure, with focal diminished and thickened Z-Bands. (**f**) Mitochondria are located in aggregates with swollen and dissolved cristae between myofibrils with occurrence of lipofuscin (Magnification (**a**–**c**) x100, (**d**) x400, (**e**,**f**) x12,000).

**Figure 2 jpm-13-00611-f002:**
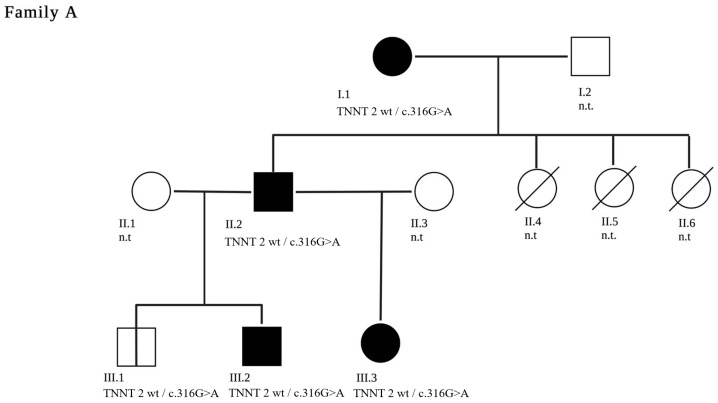
A three-generation family tree of family A is shown. The index patient is A.III.3. The clinically affected patients are marked in black. Patients A.II.4, A.II.5 and A.II.6 are deceased. The c.316G>A pathogenic variant in *TNNT2* segregated with disease in the genotyped individuals. Family members who have not been genetically tested are labelled n.t—“not tested”.

**Figure 3 jpm-13-00611-f003:**
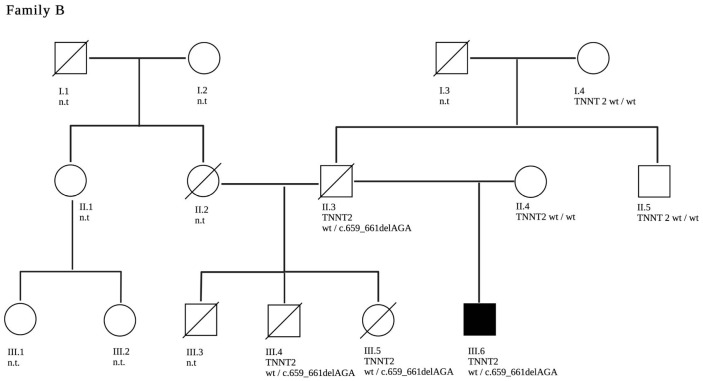
A three-generation family tree of family B is shown. The index patient is B.III.6. The clinically affected patients are marked in black. Deceased are B.II.3, B.III.3 and B.III.4. The c.659_661delAGA pathogenic variant in *TNNT2* segregated with disease in the genotyped individuals. Family members who have not been genetically tested are labelled n.t—“not tested”.

**Figure 4 jpm-13-00611-f004:**
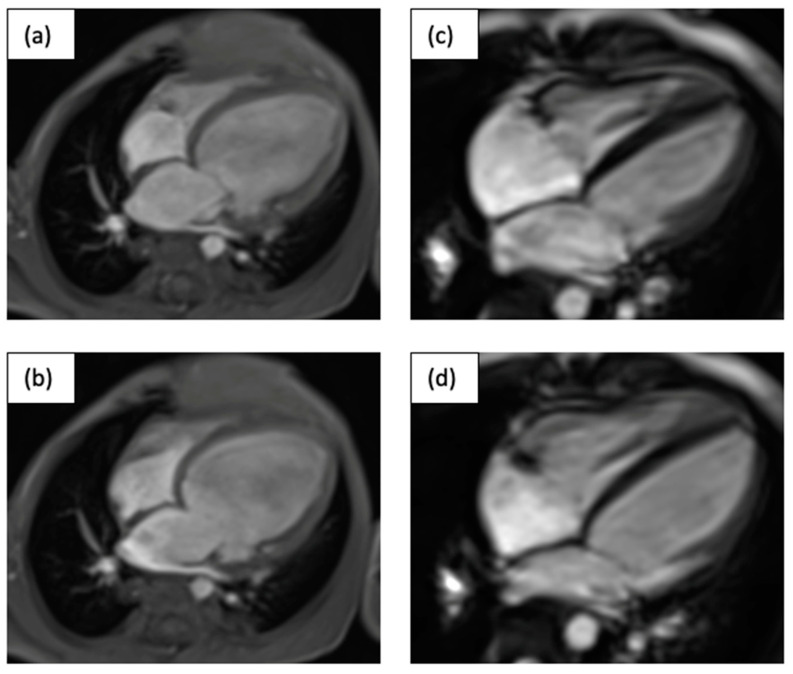
Depicted are systolic (**a**,**c**) and diastolic (**b**,**d**) MRI four-chamber views of the 17-month-old child (C.II.1) before (**a**,**b**) and 10 years after PAB (**c**,**d**) in systole and diastole.

**Figure 5 jpm-13-00611-f005:**
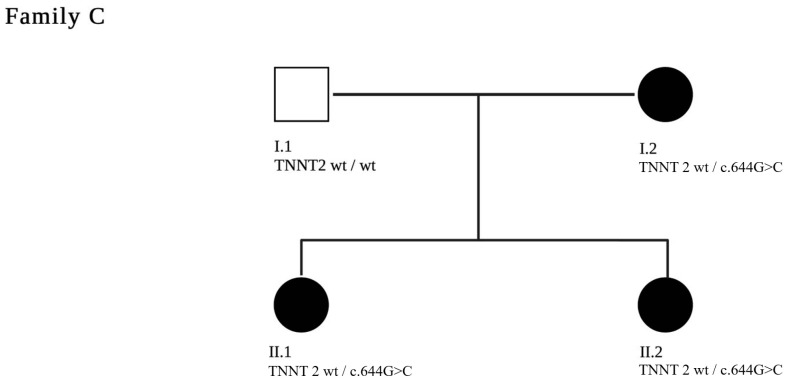
A two-generation pedigree of family C is shown. The index patient is C.II.1. The clinically affected patients are marked in black. The c.644G>C pathogenic variant in *TNNT2* segregated with disease in the genotyped individuals.

**Table 1 jpm-13-00611-t001:** Patient characteristics.

		First Admission				Follow up at Last Contact, #, &			
	Gender	Age	LVEDD (z-Score)	EF (%)	BNP (pg/mL)	Age	LVEDD (z-Score)	EF (%)	BNP (pg/mL)
A.III.3	female	2 y	+7.2	severely reduced	6528	# at the age of 2 y	#	#	#
A.III.2	male	0 m	+2.2	normal	206	# at the age of 13 y	#	#	#
A.III.1	male	6 m	−0.9	64	n.a.	5 y	+3.9	66	n.a.
B. II.3	male	n.a.	n.a.	n.a.	n.a.	& at the age 50 y	&	&	&
B.III.3	male	5 m	+8.4	reduced	n.a.	# at the age of 8 m, & at the age of 12 y	&	&	&
B.III.4	male	12 y	7.7	severely reduced	n.a.	& at the age 12 y	&	&	&
B.III.5	female	10 y	1.5	mildly reduced	n.a.	# at the age of 14 y, & at the age of 20 y	&	&	&
B.III.6	male	21 m	−0.4	normal	n.a.	12 y	+1.0	51	<50
C.II.1	female	4 y	+4.6	25	3651	12 y	+4.6	48	59
C.II.2	female	14 y	+5.1	62	n.a.	16 y	+4.8	48	29
C.I.2	female	35 y	n.a.	n.a.	n.a.	39 y	+5.2	n.a.	n.a.

EF—ejection fraction; LVEDD—left ventricular end diastolic diameter; m—months; y—years. #—patient was heart transplanted, &—patient died.

## Data Availability

The datasets used and analyzed during this study are available from the corresponding author on reasonable request.

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
