# Peer review of "Genetic Screening Reveals Heterogeneous Clinical Phenotypes in Patients with Dilated Cardiomyopathy and Troponin T2 Variants"

_jpm, 2023, doi:10.3390/jpm13040611_

Round 1

Reviewer 1 Report

This is an interesting article that describes 3 families with children diagnosed of dilated cardiomyopathy (DCM) and with a TNNT2 gene mutation.  These are families with an extreme phenotype that required heart transplant and with many deceased members. 

As for comments:

1) Family A's mutation is not fully described (only the coding change is provided).  Please provide a full description of the mutation with changes in the protein and reference transcript identification (NM number).  

2) Reference transcripts are also missing in families B and C

3) The discussion would benefit of a review of the literature describing other familiar cases with a TNNT2 mutation and the genotype-phenotype correlation.  This could be in the form of a table with found mutations and phenotypes.

Author Response

We would like to thank the reviewers and the editor for their suggestions to improve our manuscript.

COMMENTS TO Reviewer 1:

1) Family A's mutation is not fully described (only the coding change is provided).  Please provide a full description of the mutation with changes in the protein and reference transcript identification (NM number).  

The variants are fully described in the revised version of the manuscript. For family A, this corresponds to NM_001276345.2(TNNT2):c.286G>A p.(Asp96Asn).

2) Reference transcripts are also missing in families B and C

The Variants are fully described in the revised version of the manuscript. For Family B, this corresponds to NM_001276345.2(TNNT2):c.659_661del p.(Lys220del). For family C, this corresponds to NM_001276345.2(TNNT2):c.644G>C p.(Arg215Pro). In the previous version of the manuscript, the protein consequence in family C was given based on the transcript NM_000364.4 resulting in the aminoacid change p.(Arg212Pro). In order to unify the annotations throughout the manuscript, this has been extended with the position on the canonical transcript.

3) The discussion would benefit of a review of the literature describing other familiar cases with a TNNT2 mutation and the genotype-phenotype correlation.  This could be in the form of a table with found mutations and phenotypes.

The discussion was extended to the reviewer’s suggestion. A supplemental table was included with (likely) pathogenic variants in  TNNT2 as listed in Clinvar (Supplementary table 1). A supplemental figure was included to visualize the variants and the discussion of the manuscript was extended (page 8, line 242ff).

Reviewer 2 Report

The authors describe three families presenting with dilated cardiomyopathy and pathogenic variants in the TNNT2 gene. The focus of the article is on the clinical heterogeneity of the disease, showing high intrafamilial heterogeneity. Although this is not a novel finding the description of these families is interesting and merits to be published. However, the paper should be improved to be accepted for publication.

1 – The article needs extensive editing of English language and style. It should be revised by a native English speaker.

2 – All gene names should be italicized throughout the paper.

3 – The drawing of pedigrees is not correct and should be revised according to standard recommendations:

BENNETT RL et al. Standardized human pedigree nomenclature: update and assessment of the recommendations of the National Society of Genetic Counselors. J Genet Counsel (2008) 17:424-433.

BENNETT RL. The family medical history as a tool in preconception consultation. J Community Genet (2012) 3:175–183.

4 – The title of the paper should be revised. I suggest the following changes:

Genetic screening reveals heterogeneous clinical phenotypes in patients with dilated cardiomyopathy and Troponin T2 variants.

5 – Since the age of onset is largely variable, the authors should not state the penetrance is 94%, because the only individual harboring a pathogenic variant in the TNNT2 gene was 5 years old at the last evaluation.

6 – Regarding pathogenic variants in the TNNT2 gene, the authors should include which ACMG criteria were fulfilled for this classification. Also, have these variants been described previously? Are there novel variants? It should be described in the article.

7 – In the discussion section, the authors should compare there results with previous familial cases of the literature, with variants in the same gene.

Author Response

We would like to thank the reviewers and the editor for their suggestions to improve our manuscript.

Comments to Reviewer #2:

1 – The article needs extensive editing of English language and style. It should be revised by a native English speaker.

The manuscript was edited accordingly.

2 – All gene names should be italicized throughout the paper.

All gene names were italicized.

3 – The drawing of pedigrees is not correct and should be revised according to standard recommendations:

BENNETT RL et al. Standardized human pedigree nomenclature: update and assessment of the recommendations of the National Society of Genetic Counselors. J Genet Counsel (2008) 17:424-433.

BENNETT RL. The family medical history as a tool in preconception consultation. J Community Genet (2012) 3:175–183.

The Pedigrees were changed and drawn according to standard recommendation (Bennett et al.)

4 – The title of the paper should be revised. I suggest the following changes:

Genetic screening reveals heterogeneous clinical phenotypes in patients with dilated cardiomyopathy and Troponin T2 variants.

The title of the manuscript was changed according to the reviewer’s suggestion.

5 – Since the age of onset is largely variable, the authors should not state the penetrance is 94%, because the only individual harboring a pathogenic variant in the TNNT2 gene was 5 years old at the last evaluation.

The manuscript was changed according to the reviewer’s suggestion.

In our patient cohort with variants in the TNNT2, primary DCM occurred as a familial form with autosomal dominant inheritance and penetrance of 94% or higher as the patients without penetrance were at younger age at the actual follow up….

6 – Regarding pathogenic variants in the TNNT2 gene, the authors should include which ACMG criteria were fulfilled for this classification. Also, have these variants been described previously? Are there novel variants? It should be described in the article.

The classification in accordance to the ACMG criteria has been added in the supplement (Supplementary table 2). The variants in family A and family B have previously been reported. Our manuscript extends these information by adding phenotypic and segregational data across a multigenerational pedigree. To our knowledge, the variant found in family C has not yet been reported.

7 – In the discussion section, the authors should compare there results with previous familial cases of the literature, with variants in the same gene.

The discussion was extended to the reviewer’s suggestion. A supplemental table was included with variants of TNNT2 described so far with the use of Clinvar. A supplemental figure was included to visualize the variants and the discussion of the manuscript was extended (page 8, line 242ff).

Reviewer 3 Report

The study titled "Genetic screening reveals heterogeneous clinical causes in patients with DCM and Troponin T2 variants" by Angelika et al., is relevant to the field in identifying and treating cardiomyopathies (CM). A comprehensive approach is needed for identification and treatment of CM owing to the genetic variations and other unknown causes. This report will add to the identification of genetic variants prevalent in the etiology of CM. The study is very clear and carried out aptly. However, in this report it would be useful to discuss whether these mutations identified were reported in any other similar etiology of CM and may have any similarity or differences with phenotypic characteristics and outcomes if earlier reported to strengthen the value of this finding. 

Author Response

We would like to thank the reviewers and the editor for their suggestions to improve our manuscript.

Comments to Reviewer #3:

The study is very clear and carried out aptly. However, in this report it would be useful to discuss whether these mutations identified were reported in any other similar etiology of CM and may have any similarity or differences with phenotypic characteristics and outcomes if earlier reported to strengthen the value of this finding. 

The mutation identified and further mutations in the TNNT2 gene were discussed in more detail (page 8, line 242ff). Additionally, a supplementary figure was added to visualize the position of the mutation and a table was added with pathogenic and likely pathogenic mutations in the TNNT2 gene.

Round 2

Reviewer 2 Report

The manuscript has been improved. However , some suggestion were not fully addressed:

1 - The drawing of pedigrees is still not correct and should be revised. Union lines should be between two individuals, and not below. Generations should be numbered with Roman numerals. Descendance lines should be straight and not diagonal. Please, revised the pedigrees according to standard recommendations.

2 – The description of the variants in the discussion, which have been previously described, which are novel variants, is not sufficiently well described. Please, revise this paragraph of the discussion section, highlighting this aspect. The same applies for comparison of the results with previous familial cases of the literature.

Author Response

We would like thank the reviewer for his comments.

1 - The drawing of pedigrees is still not correct and should be revised. Union lines should be between two individuals, and not below. Generations should be numbered with Roman numerals. Descendance lines should be straight and not diagonal. Please, revised the pedigrees according to standard recommendations.

The pedigrees were revised according to the reviewer’s suggestion and the standard recommendation. 

2 – The description of the variants in the discussion, which have been previously described, which are novel variants, is not sufficiently well described. Please, revise this paragraph of the discussion section, highlighting this aspect. The same applies for comparison of the results with previous familial cases of the literature.

During the revision we realized and internal miscommunication. Due to a miscommunication the cDNA change in family A was converted to the canonical transcript based on a false initial transcript. Following the annotation in family C, NM_000364.4 was assumed, although the appropriate transcript was NM_001001430.3. This lead to a false annotation of the variant from family A in the revised transcript. After internal review of this issue, we have corrected the corresponding sections of the main text and the supplement. We have included a new part in the discussion (p8 l 219-p9 l 268) and extended the supplement.